# Spotted Fever Group *Rickettsia* spp. Molecular and Serological Evidence among Colombian Vectors and Animal Hosts: A Historical Review

**DOI:** 10.3390/insects15030170

**Published:** 2024-03-02

**Authors:** Lídia Gual-Gonzalez, Myriam E. Torres, Stella C. W. Self, Omar Cantillo-Barraza, Melissa S. Nolan

**Affiliations:** 1Department of Epidemiology and Biostatistics, Arnold School of Public Health, University of South Carolina, Columbia, SC 29208, USA; lidiag@email.sc.edu (L.G.-G.); torresme@mailbox.sc.edu (M.E.T.); scwatson@mailbox.sc.edu (S.C.W.S.); 2Biology and Infectious Disease Control Group, University of Antioquia, Medellin 050010, Colombia; omar.cantillo@udea.edu.co

**Keywords:** spotted fever group rickettsia, Colombia, rickettsiosis, tick-borne diseases, one health

## Abstract

**Simple Summary:**

Colombia is one of the countries most affected by spotted fever group *Rickettsia* spp. (SFGR) in Latin America, yet these infections are not nationally reportable. Diagnostic challenges and lack of vector surveillance are likely contributing to an understudy of these infections. This literature review aims to bring together the research articles related to SFGR in Colombia dating back to its first description in 1935. This review aims to identify areas for clinical management improvement and promote prospective SFGR investigation in Colombia.

**Abstract:**

Spotted fever group *Rickettsia* spp. (SFGR) are a large group of tick-borne bacteria causing important emerging and re-emerging diseases that affect animals and humans. While SFGR are found worldwide, a lack of surveillance and misdiagnosis particularly affect South American countries. Colombia is a high burdened country in South America, yet rickettsioses are not deemed a nationally reportable condition limiting disease-specific public health resources. As mortality rates are high for one *Rickettsia* pathogen species, there is a great need to better understand the epidemiological and ecological factors that increase SFGR transmission risk regionally. This literature review provides an overview of Colombia-based SFGR studies connecting knowledge about both vectors and hosts.

## 1. Introduction

*Rickettsia* spp. within the spotted fever group (SFGR) are emerging tick-borne bacteria that can cause disease, and if left untreated, it can progress to severe complications and death [1,2]. While *Rickettsia rickettsii*, the causal agent of Rocky Mountain spotted fever (RMSF), is recognized as the most severe infection, other spotted fever group (SFG) rickettsioses have gained importance over the past decades [2]. SFG rickettsiosis treatment is widely available, but obtaining a timely diagnosis remains challenging, which contributes to SFG rickettsiosis’ status as a neglected disease [3]. Diagnosis involves assessing clinical and epidemiological evidence, and the utility of laboratory diagnostics is limited: microscopic and molecular techniques have a short timeframe to be performed due to the intracellular nature of the bacteria, and serologic techniques will often take weeks to complete as patients seroconvert after 5–7 days post-infection, which would cause treatment delay, increasing the risk of severe complications [4]. 

SFGR are highly present in South American countries—countries which often lack adequate surveillance infrastructure [1]. Colombia is a high burdened country in South America yet rickettsioses are not a nationally reportable disease, yielding an underestimation of the actual disease burden. Moreover, Colombia is a South American country with high fauna biodiversity, where SFGR have ample host reservoirs for pathogen life-cycle maintenance [5]. Therefore, epidemiological surveillance should rely not only in human populations but should be complemented by monitoring tick vectors as well [6]. For this reason, this review summarizes what is known about these existing transmission dynamics in Colombia, aiming to become a catalyst for improved research and surveillance nationwide. This literature review describes important tick species in Colombia and their potential to vector SFGR, the different SFGR species described up to date, and the epidemiological implications among humans and animals.

## 2. Clinical Manifestations and Management

In Colombia, two pathogenic SFGR species have been identified: *Rickettsia rickettsii*, and *Rickettsia parkeri* [7,8]. *Rickettsia rickettsii* causes RMSF, also known as “Tobia fever” in Colombia and “Brazilian Spotted fever” in Brazil [9,10]. RMSF clinical presentation often presents early unspecific signs including fever and headache, and it rarely comes accompanied by inoculation eschar. Between days 2 and 4, a macular rash appears in 50–60% of cases around the wrists and ankles with some patients developing cough, nausea, vomiting, diarrhea, abdominal pain, and anorexia. If the patient is left untreated, disease progresses: around days 5–7, post-bite fever increases (>40 °C), respiratory symptoms and abdominal pain worsens, and the macular rash develops into generalized petechiae in the hands and feet. Around days 7–9, the late-stage rash progresses to a diffuse eruption across the entire body, peripheral gangrene and necrosis, septic shock, myocarditis, arrythmia, kidney failure, pulmonary edema, meningoencephalitis, and death [11,12]. *Rickettsia parkeri* rickettsiosis is considered the second most common SFGR: in Latin America, the strain Atlantic Rainforest is predominant [7,13,14]. *Rickettsia parkeri* infection presents with unspecific symptoms similar to other tick-borne rickettsiosis with a majority of patients presenting with fever (94%) and an eschar (91%) and less presenting with a rash (72%) [14]. The less frequent symptoms overlap with RMSF with patients showing headaches, myalgias, malaise, arthralgias, and chills. 

Among other SFGR species, *Rickettsia amblyommatis* has been debated to be associated with mild disease. A tick removed from a woman in North Carolina, USA, that developed a rash at the tick bite site yielded a solo *R. amblyommatis* diagnosis, suggesting a possible association between this species and mild symptoms [15]. Similarly, another species has been suspected to cause clinical disease: in 2002, an outbreak in Perú found evidence of a novel *Rickettsia* sp. associated with febrile illness [16]; this species was later proposed to be named “*Candidatus* Rickettsia andeanae” [17]. The symptoms described among the patients diagnosed with SFGR included fever and headache, 75% of them reported malaise, and 50% or less showed other symptoms overlapping RMSF. Clinical manifestation associated with *R. amblyommatis* and *Candidatus*
*R. andeanae* has been rarely found, and additional evidence is needed to confirm the pathogenicity of these two species.

Clinical management is highly effective if provided early; doxycycline has been shown to be effective to treat in patients that are suspected to have SFGR infections. Fatal cases are often associated with delayed diagnosis; thus, treatment is recommended when physicians suspect patients are infected with SFGR [12]. In addition, chloramphenicol is recommended for pregnant women and tetracyclines in case of life-threatening situations during pregnancy.

## 3. Molecular Evidence

In Colombia, ticks have been described and re-described yielding 67 species identified within eight genera: *Argas*, *Antricola*, *Ornithodoros*, *Ixodes*, *Rhipicephalus*, *Dermacentor*, *Amblyomma*, and *Haemaphysalis* [18,19,20]. Thirteen of these species have been found infected with *Rickettsia* spp. within the SFGR assemblage (Table 1). During the 20th century, Colombian tick descriptions relied on morphological characterization. Later inventions of molecular techniques allowed the redescription of previously classified tick species as species complexes, including the characterization of the species *Amblyomma patinoi* and *Amblyomma mixtum* from the *Amblyomma cajennense* sensu lato complex [21]. The literature has documented five SFGR species infecting these ticks nationally.

### 3.1. Rickettsia rickettsii

Evidence of the most pathogenic species among the spotted fever group, *R. rickettsii*, originated in the 1940s following the first described outbreak associated with *A. cajennense* s.l. ticks in Tobia [9]. *A. cajennense* s.l. are considered important *R. rickettsii* vector species in South America, but recent advances discovered this species forms a complex of which species *A. patinoi* and *A. mixtum* are found in Colombia [28,40]. Since then, *R. rickettsii* has been detected in *A. patinoi*, *A. mixtum* and *D. nitens*, which are all considered the primary vector species in Colombia (Table 1).

*Rickettsia rickettsii* infection rates in *A. patinoi* have been described at around 6.6% in Colombia, and around 2.0% in *A. cajennense* s.l., 1.0% in *Amblyomma* sp., and 10.0% in *D. nitens* [25]. Similar rates are described in Brazil: 0.5% prevalence was described in *A. cajennense* s.l., whereas between 0.05 and 11.66% was described in *R. sanguineus* s.l., and around 0.89% was described in *A. aureolatum* [41]. *Rhipicephalus sanguineus* s.l. has been evaluated in a couple studies in Colombia and has been found negative for *R. rickettsii* [42,43]. Infection rates have been found higher (20.7% prevalence) among *A. mixtum* collected horses [23]. The higher prevalence could be explained by co-feeding with other infected ticks [44], or associated with the particular tick species, which is potentially one of the primary vectors of *R. rickettsii* in Colombia [41]. Domestic animals are considered responsible for bringing ticks closer to humans; however, despite being an important epidemiological risk factor, domesticated animals are still overlooked [45]. 

Identifying cases molecularly is challenging because SFGR bacteria systemically circulate in the bloodstream for a short period of time, although species detection through PCR can be performed from an eschar swab [46]. Given the low presence of eschar in RMSF patients, clinical and epidemiological assessment is the current diagnostic method. Diagnosis nationally is hindered by physician unawareness, which is one of the primary barriers in obtaining diagnosis and subsequent treatment [3]. Rickettsioses are not nationally reportable; underreporting of *R. rickettsii* cases and an unknown distribution of vectors contribute to unawareness of SFGR risk among physicians. Failure to recognize and treat cases contributes to increased mortality, which is estimated to be somewhere between 26–54% [8,47]. Vector species most likely associated with human transmission are *A. patinoi* and *D. nitens*, as these are known anthropophilic species in South America [37], although it is possible that *A. mixtum* can bite humans [48].

### 3.2. Rickettsia parkeri

Another SFGR species described in Colombia is *R. parkeri*. This pathogenic species was first described in 2010, and the strain Atlantic Rainforest has become more important in South America given its link to human cases [49]. This strain has been detected in Colombia among *A. ovale* [13] and *A. nodosum* ticks [34]. A 3.33% infection rate was described in *A. nodosum* collected from birds, which is a tick often found associated with passerine birds. On the other hand, Londoño et al. found an approximately 12.7% *R. parkeri* infection rate among *A. ovale* [50]. *Amblyomma ovale* were collected mostly on dogs, and considering the anthropophilic nature of this tick species, *R. parkeri* was considered an exposure risk in the area [36]. The authors hypothesized that the population could be gaining cross-immunity against *R. rickettsii* from mild *R. parkeri* infections [50]. Later, a case report described the first human case in Antioquia, Colombia: a male farmer with eschar and draining lymphadenopathy who developed minimal clinical symptoms; this case was successfully treated and characterized through molecular diagnosis [7]. Despite the geographical distribution of potentially infected ticks, only one human *R. parkeri* case report has been documented in Colombia [7], thus warranting further investigations in order to unravel the risk of this pathogenic species.

### 3.3. Rickettsia amblyommatis

*Rickettsia amblyommatis* has gained attention in recent years due to its suspected pathogenicity [51]. Some authors hypothesize that this species’ pathogenicity might be different across strains, whereas others suggest it has a role as a tick endosymbiont, which could explain its wide geographical distribution [52,53]. In Colombia, *R. amblyommatis* has been detected molecularly among *A. cajennense* s.l. and *R. microplus*, including a described strain *R. amblyommatis* strain “Conduru” [25,29]. Multiple tick-infested animal surveys have detected the presence of *R. amblyommatis* in *A. longirostre*, *A. varium*, *R. microplus*, and an unidentified *Ixodes* sp., highlighting the widespread tick–host suitability for *R. amblyommatis* [25,31,36]. In 2023, a study in the Orinoquia region determined the presence of *R. amblyommatis* in *A. mixtum* for the first time in ticks collected from the vegetation [33]. Knowledge of *R. amblyommatis*’ host reservoirs remains limited as it is difficult to confirm *R. amblyommatis* through indirect fluorescent antibody (IFA) assay slides. Veritable *R. amblyommatis* infection could be misdiagnosed with other *Rickettsia* spp. given the cross-reactivity on clinically available IFA slides [25,42,54]. Despite the historical belief that this species is an endosymbiont found in *Amblyomma* sp. ticks, recent evidence of this species in other tick genera has motivated studies considering its pathogenicity or its role in enhancing immunity against other virulent *Rickettsia* spp. [51,52].

### 3.4. Candidatus Rickettsia colombianensi

In 2012, a study provided evidence of a novel species *Candidatus*
*R. colombianensi*, which is a species found in *A. dissimile* ticks from northern Colombia [24]. First described in ticks collected from iguanas in the Córdoba department, this emerging species has been described across north and central Colombia (Figure 1) [24,27,30,55,56,57]. In its first description, 24.7% of *A. dissimile* pools showed the presence of *Rickettsia* sp. Authors then isolated *Candidatus*
*R. colombianensi* from four of the positive pools [24]. *Candidatus*
*R. colombianensi* pathogenicity has not been described, and high infection rates are noted among *A. dissimile*, where more than half of the adult, nymph and larvae pools are positive for at least one *Rickettsia* sp., including *Candidatus*
*R. colombianensi*, which could imply humans might be highly exposed to this species [55,58]. *Amblyomma dissimile* infection with *Candidatus*
*R. colombianensi* is described in several studies across the country, which is likely associated with the host ticks’ bloodmeal preference for amphibians and reptiles [23,30,31,36,59,60]. *Candidatus*
*R. colombianensi* has only been identified from sampling areas with humid or semi-humid climates; thus, increased surveillance in other climates is warranted to understand the ecology of this species. While the presence of *A. dissimile* in Colombia is primarily associated with reptiles and birds [24,56], it might occasionally play a role in domestic transmission: for example, horses with attached *A. dissimile* ticks have been reported positive for anti-SFGR antibodies, and some of the collected ticks showed the presence of *Candidatus*
*R. colombianensi* [38]. This species gained relevance when it was found among anthropophilic ticks within the *A. cajennense* s.l. complex, which are relevant for human SFGR epidemiology in Colombia [40]. One study collected host-attached *A. cajennense* s.l. among capybaras and found that 2.6% were positive for *Candidatus*
*R. colombianensi* [27]. Among the *A. cajennense* complex, *Candidatus*
*R. colombianensi* has also been found in *A. mixtum*, which is a known human-biting tick; thus, implications of this species in SFGR epidemiology should be further evaluated [23]. 

### 3.5. Candidatus Rickettsia andeanae

The last SFGR that has been reported in Colombia is *Candidatus*
*R. andeanae*. This species has been described in *A. ovale*, *A. maculatum* and *R. sanguineus* s.l. [23]. Authors reported a 3.8% positivity rate in *R. sanguineus* s.l., 40% in *A. ovale*, and 15.8% in *A. maculatum.* The ticks collected were attached to dogs, which has been suggested to be an important risk factor in human exposure [61]. Given that these ticks are known to have anthropophilic behavior, it is likely that in Colombia, humans can be exposed to *Candidatus*
*R. andeanae* in some areas [37]. Although there is no evidence this species is associated with human pathogenicity, some studies suggest it could play an antagonistic role against *R. parkeri*, which is a known pathogenic species [62]. This would have important implications for SFGR epidemiology, such as lower pathogenic species exposure or cross-immunity by previous exposure to non-pathogenic *Rickettsia* species.

## 4. Serological Evidence

Historical records document the first reported Colombian SFG rickettsiosis outbreak in 1935. Cases from an unknown disease, named “Tobia fever”, happened repeatedly in the Villeta municipality, Cundinamarca department within the same households, with a 20% incidence and 95% fatality rate [9]. It was later determined that *R. rickettsii* was the causative agent, although no other investigations or cases were reported in Colombia until 2001 [63]. Several outbreaks have occurred since then (Table 2). In 2007, two fatal *R. rickettsii* cases were reported in Villeta, which led to a serosurveillance investigation that found 4.7% seropositivity at the national level and 21.9% seropositivity at the regional level [64]. No more studies have used national data, and given rickettsioses are not reportable, there are no current seroprevalence estimates across Colombia. Lack of surveillance turns SFG rickettsioses into neglected diseases, contributes to physician unawareness, and increases the risk of fatal cases due to misdiagnosis [8]. There is a need for passive surveillance and improved education to reduce misdiagnosis and increase treatment administration.

Human SFG rickettsioses depend on animal populations to propagate transmission in sylvatic and peridomestic environments. Domestic animals, particularly dogs, serve as transmission bridges by bringing infected ticks closer to humans and by serving as an alternative bacteremic source for naïve domestic ticks. Surveillance and serological studies involving animals can be useful to understand transmission dynamics. The reported seroprevalence in domestic animals has been mainly performed in the northwestern regions in Antioquia and Cordoba, which conveniently are also regions were most human outbreaks and human studies are performed [43,50,69]. Studies have found that a higher infection risk is associated with living close to other positive individuals, and owning older livestock, but inconsistent methods hinder the external validity of the results [25,50,74].

Considering the animal host diversity around outbreak regions, studies have also implicated several sylvatic hosts as possible SFGR reservoirs. Birds have been studied, given their migration patterns and travel distances, and could be considered important reservoir hosts [31,77,79,80]. Although limited evidence exists on SFG rickettsiosis prevalence among birds, knowledge on tick infestation rates and species associated with birds may yield some information on possible enzootic cycles [80]. Peridomestic mammals such as capybaras and opossums that live near communities are also considered important hosts linked with infection [39,70]. Opossums have been found associated with a greater likelihood of tick exposure among humans [39]. Alternatively, capybaras may serve as possible *R. rickettsii* amplifying hosts, as their rickettsemia lasts longer without causing clinically apparent disease, differing from other sylvatic hosts that are susceptible to illness [81,82]. 

Indirect fluorescent antibody (IFA) test slides are commonly used in epidemiological studies; however, due to cross-reactivity, species confirmation cannot be obtained. Different species could explain differences in seroprevalence estimates and associated factors: for example, two studies in rural and indigenous communities reported a 29.2% IgG seropositivity in Antioquia and 32% seropositivity in Cesar [68,72]. Both studies showed seropositivity was associated with being female, a housewife occupation, and younger age. Conversely, studies in different rural communities found seropositivity (ranging between 20.4% and 26.7%) was associated with being male, having domestic animals, living close to wildlife, having greater number of domestic animals, or outdoor worker occupation [36,42,67,78]. A study among only male farmers reported a 52.1% seropositivity, suggesting this occupational group is at highest risk, which is likely due to exposures associated with their job responsibilities [74]. Given the wide geographic range of vectors, vertebrate hosts, and associated factors, increased efforts should be given to understand the transmission patterns and risk factors in Colombia to mitigate SFGR infection spread.

## 5. Vector Competence

Vector competence of the Colombian tick species has only been described using *R. rickettsii,* which can be used to understand the transmission dynamics of this bacterial group. *A. patinoi* is considered the primary *R. rickettsii* vector [28,39], and this species has demonstrated a successful maintenance of *R. rickettsii* through transstadial perpetuation, transovarial transmission, and subsequent pathogen transmission post-colony passage perpetuation [83]. Despite this evidence, some authors argue that this vector is less susceptible to *R. rickettsii* infection compared to other vectors present in Colombia, which is likely due to the higher presence of non-pathogenic species among this vector [84]. Among the species complex, some suggestions point out that *A. mixtum* could be considered an important *R. rickettsii* vector [32]. 

Other important *R. rickettsii* vectors have been observed in Colombia: *R. sanguineus* and *D. nitens*. *R. sanguineus* has demonstrated successful maintenance of *R. rickettsii* through transstadial perpetuation and an effective infection of susceptible hosts; however, this vector has not been found infected with *R. rickettsii* in Colombia, suggesting possible microbiome composition interactions affecting the pathogen distribution [23,25,36,42,43,44,45,85]. *Dermacentor nitens*, on the other hand, are considered a probable vector, yet its competence has not been demonstrated [86]; despite being found infected with *R. rickettsii* in Colombia, its importance in transmission remains unknown [25]. 

In the case of the *R. parkeri* strain Atlantic Rainforest, it has been found infecting *A. ovale* and *A. nodosum* ticks in Colombia [13,23,34,36]. *Amblyomma ovale* has been demonstrated to have vector capacity [87] and to maintain *R. parkeri* strain Atlantic Rainforest transstadially and transovarially under laboratory conditions for two generations [88]. Conversely, *A. nodosum,* a suspected alternative possible vector, is not known to feed on human populations [37]; therefore, its implications in human transmission remain unclear.

*Amblyomma maculatum* is associated with *R. parkeri* in the United States of America [89], but this tick species has only been found infected with *Candidatus*
*R. andeanae* in Colombia. It is hypothesized that *A. maculatum* infected with *Candidatus*
*R. andeanae* are refractory to *R. parkeri* infection, thus reducing vector capacity [90]. Exclusion relationships between *Rickettsia* spp. remain understudied, and future studies should evaluate other vector and SFGR species’ biological interactions to understand transmission dynamics and vector competence consequences.

## 6. Conclusions

Literature reviews focusing on SFG *Rickettsia* sp. are scarce and are necessary to improve our understanding of this group of neglected infections. This review focuses on the SFG *Rickettsia* sp. evidence in Colombia. Since the first outbreak in the 20th century, five SFGR species have described in Colombia: *R. rickettsii*, *R. parkeri* Strain Atlantic Rainforest, *R. amblyommatis*, *Candidatus*
*R. colombianensi*, and *Candidatus*
*R. andeanae*. Only *R. rickettsii* and *R. parkeri* are pathogenic species in Colombia; human exposure to other species could be influencing the epidemiology of rickettsiosis, warranting a need for increased surveillance [25,36]. Thirteen tick species have been reported to be infected with SFGR in Colombia. Most tick species are commonly found infesting wild fauna or domestic animals [31,55,79,85], and *A. cajennense* s.l., *A. patinoi*, *A. mixtum*, and *A. ovale* are likely the primary SFGR human tick vectors in Colombia. 

Despite the evidence of highly pathogenic SFGR species in Colombia, rickettsiosis remain a non-reportable disease, hindering clinical management and diagnosis [91]. Given the high prevalence of other vector-borne diseases nationally, it is important to have an accurate epidemiological context of SFG rickettsioses to ensure early treatment is provided to suspected cases. Physician unawareness leads to misdiagnosis and preventable deaths; therefore, improved studies are warranted to better understand the distribution of these pathogenic species [3]. Lastly, a better understanding of the intricate associations between the different vectors and *Rickettsia* species would afford scientists to complete the epidemiological understanding of these public health important infections.

## Figures and Tables

**Figure 1 insects-15-00170-f001:**
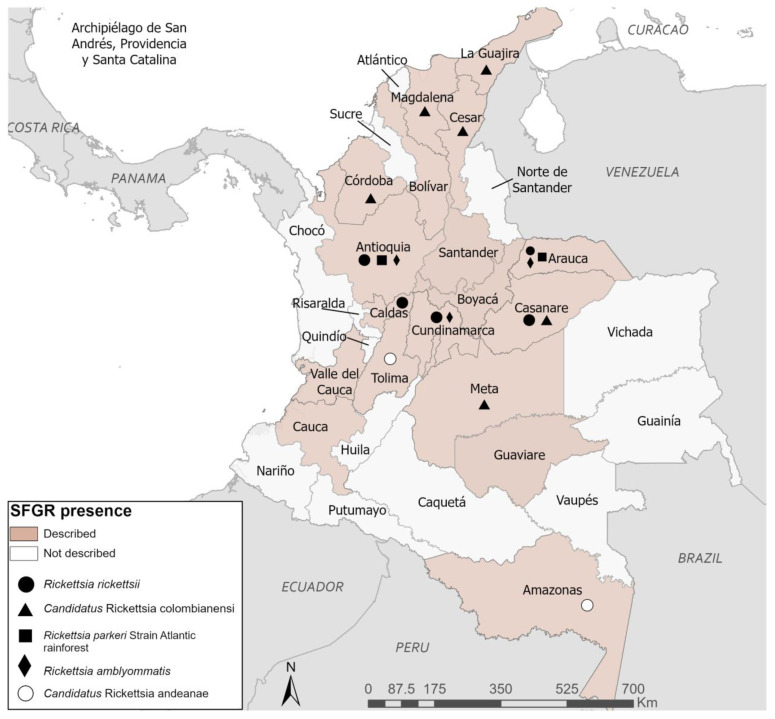
Spotted fever group *Rickettsia* species molecularly identified and serological evidence of SFGR across Colombia.

**Table 1 insects-15-00170-t001:** Colombian ticks, probable hosts, spotted fever group *Rickettsia* spp. associated and year it was first described in Colombia.

Ticks (Acari: Ixodida)	Hosts Order	*Rickettsia* Species	Year	References
*Rhipicephalus sanguineus* s.l. (Latreille, 1806)	Canidae, humans	*Candidatus* *R. andeanae* ^§^	1929	[22,23]
*Rhipicephalus microplus* (Canestrini, 1888)	Artiodactyla, Canidae, Aves, humans	*Rickettsia* sp., *R. amblyommatis*, *Candidatus* Rickettsia colombianensi ^§^	1928	[22,24,25]
*Dermacentor nitens* (Neumann, 1897)	Perissodactyla, Artiodactyla, humans	*R. rickettsii*	1936	[18,25]
*Amblyomma cajennense* s.l. * (Fabricius, 1787)	Artiodactula, Canidae, Cingulata, Didelphimorphia, Perissodactyla, Pilosa, Rodentia, humans	*Rickettsia* sp., *R. rickettsii*, *R. amblyommatis*, *Candidatus* *R. colombianensi* ^§^	1899	[25,26,27,28,29]
*Amblyomma dissimile* (Koch, 1844)	Squamata, Anura, Aves, humans	*Candidatus* *R. colombianensi* ^§^	1899	[18,24,30,31]
*Amblyomma longirostre* (Koch, 1844)	Rodentia, Aves	*R. amblyommatis*	1923	[18,26,31]
*Amblyomma maculatum* (Koch, 1844)	Perissodactyla, Artiodactyla, Carnivora, Aves	*Candidatus* *R. andeanae* ^§^	1919	[18,23,26]
*Amblyomma mixtum* * (Koch, 1844)	Perissodactyla, Artiodactyla, Rodentia, Aves	*R. rickettsii*, *R. amblyommatis*	2016	[23,32,33]
*Amblyomma nodosum* (Neumann, 1899)	Pilosa, Aves	*R. parkeri*	1942	[18,34]
*Amblyomma ovale* (Koch, 1844)	Perissodactyla, Canidae, Rodentias, Didelphimorphia, Aves, humans	*Candidatus**R. andeanae*^§^, *R. parkeri*	1899	[13,23,35,36]
*Amblyomma parvum* (Argão, 1908)	Canidae, Perissodactyla	*Rickettsia* sp.	1985	[37,38]
*Amblyomma patinoi* * (Nava, Beati & Labruna, 2014)	Artiodactyla	*R. rickettsii*, *R. amblyommatis*, *Candidatus* *R. colombianensi*^§^	2013	[28,39]
*Amblyomma varium* (Koch, 1844)	Pilosa, Aves, humans	*R. amblyommatis*	1926	[18,31,36]

* Species within the same species complex. ^§^ Denotes non-pathogenic species.

**Table 2 insects-15-00170-t002:** Historical documentation of serological evidence of SFGR infection in humans or animals in Colombia.

Year	Department(s)	Diagnostic Test(s)	Antibody(ies)	Titers	Type	Reference
1935	Cundinamarca	Xenodiagnoses	N/A	N/A	Outbreak	[9]
2001	Cordoba	N/A	N/A	N/A	Outbreak	Cross-reference [63]
2003	Cundinamarca	IHA, PCR	N/A	N/A	Outbreak	[64]
	Santander, Guaviare, Caldas	IFA	IgG, IgM	1:64 to 1:1024	Surveillance	
2006	Antioquia	N/A	N/A	N/A	Outbreak	Cross-reference [65]
2007	Cundinamarca	IFA	IgG	1:64 to 1:1024	Study	[66]
	Cordoba	IFA	IgM, IgG	1:64 to 1024	Outbreak	[65]
2008	Antioquia, Cordoba	IFA	IgG	1:64 to 1:131,072	Study	[67]
2009	Antioquia	IFA	IgG	1:64 to 1:2048	Study	[68]
	Cundinamarca	IFA	IgG	1:64 to 1:8192	Study	[69]
	Cordoba	IFA	IgG	1:64 to 1:512	Study	[70]
2010–2012	Antioquia, Cordoba	IFA	IgG	1:64	Study	[50]
2011–2012	Antioquia, Cordoba	IFA	IgG	1:64	Study	[71]
2011–2013	Cundinamarca	IFA	IgG	1:64	Study	[25]
2012–2013	Cesar, Guajira	IFA	IgG	1:64	Study	[72]
2013–2014	Antioquia	IFA	IgG	1:64 to 1:512	Study	[73]
2013–2015	Antioquia	IFA	IgG	1:64 to 1024	Study	[43]
2014	Antioquia	IFA	IgM, IgG	1:16	Study	[74]
2014–2015	Antioquia	IFA	N/A	1:128 to 1:16,384	Outbreak	[45]
2015	Bogota DC	IFA	IgM	1:128 to 1:512	Case report	[75]
2015–2016	Antioquia	IFA	IgG	1:128	Study	[36]
	Antioquia	IFA	IgG	1:128	Study	[54]
2016–2018	Antioquia	IFA	IgG	1:128 to 1:8172	Study	[42]
2016–2019	Caldas	IFA	IgG	1:64 to 1:2048	Study	[76]
2018–2019	Arauca	IFA	N/A	1:64 to 1:1024	Study	[77]
2020	Antioquia	IFA	N/A	1:64 to 1:1024	Case report	[7]
2022	Boyacá	IFA	IgG	1:64 to 1:128	Study	[78]

N/A = Not Applicable. HA = Indirect hemagglutination test. PCR = Polymerase chain reaction. IFA = Indirect fluorescent antibody.

## Data Availability

No new data were created or analyzed in this study. Data sharing is not applicable to this article.

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
