# Peer review of "Spotted Fever Group Rickettsia spp. Molecular and Serological Evidence among Colombian Vectors and Animal Hosts: A Historical Review"

_insects, 2024, doi:10.3390/insects15030170_

Round 1

Reviewer 1 Report

Comments and Suggestions for Authors

Review in the file below

Author Response

Thank you so much for your comments.
The authors followed the two reviewers' suggestions, following reviewer 1 suggestions, we included a section for clinical outcomes and management.

We reduced some of the sections on molecular testing and vector competency and framed the text adding more citations to the research conducted on SFGR in other locations. We shortened the text on these sections to reduce overlapping between both paragraphs as it had some repeating parts.

We revised the text to ensure consistency, correcting Latin names and following both reviewers' suggestions.

Reviewer 2 Report

Comments and Suggestions for Authors

The manuscript ‘Spotted Fever Group Rickettsia spp. Molecular and Serological Evidence Among Colombian Vectors and Animal Hosts: A Historical Review’ is interesting but needs some revisions to make it suitable for publication. In general, there are several parts that are repeated (presented from different perspectives, but this results in a repetition: e.g. the presence of different SFG rickettsiae in different tick species, and then different tick species as hosts of different rickettsiae..), so I would revise and shorten the manuscript. Moreover, the fact that ticks are SFGR reservoir hosts, apart than vectors, should be better underlined. The topic of possible interactions among different SFG species in ticks is really interesting but it is mentioned only superficially, with various sentences scattered in the manuscript; I would suggest a specific paragraph on the topic. English language should be also revised.

In detail:

Introduction:

-revise line 29: SFG rickettsiae ‘left untreated’ – the disease can be untreated, not the bacteria

-line 35: why ‘other’ lab diagnostic?

-line 37: why serology takes so long? Please explain

-line 44: I am not sure that routine surveillance should include populations other than humans.. unless you talk about ticks.. to whom are you referring?

Table 1: ‘year’ refers to what?

-line 78-79: you say that infection rates are higher among ticks collected from animals, but you support this statement by comparing R.rickettsii prevalence in A.mixtum from horses with the prevalence in other tick species, so difference in prevalence could be linked to the vector species and not to the fact that the tick is on host or questing..

-line 84: you could mention here that PCR from eschar swabs seems a promising diagnostic approach (see for example: https://www.ncbi.nlm.nih.gov/pmc/articles/PMC3647493/)

-line 89: morbidity/mortality in the Country by R.rickettsii could be mentioned in this paragraph

-line 93: ‘bite’ instead of ‘infect’?

-line 100-103.. who are ‘the authors’?

-line 135: delete ‘even though’.. the fact that R.colombianensis is not recognized as a pathogen is not in contrast with the fact that its prevalence is high in ticks

-lines 142-144: so sampling is done mainly in humid/semi-humid areas in Colombia?

-Line 144-5: the association A.dissimile-reptiles/amphibilans/birds is already mentioned in lines 141, and again in line 251.

-line 153: the fact that R.colombianensis is not associated to disease is already said before (line 136)

-line 187: what do you mean with ‘animal studies’? Serology? Molecular? Xenodiagnosis?

-line 192: increasing animal age? Which animals?

-line 199 and following: revise. You say twice that opossums are peridomestic. Please explain better what do you mean with ‘differing from other sylvatic hosts’; capibara is the only suspected to be an amplifying host?

Chapter 4 overlaps a lot with chapter 2 in the information given.

-line 241: please provide a reference for the statement on Candidatus R. andeanae influencing R. rickettsii  maintenance in R. sanguineus

-lines 264-5: please revise, this sentence is not clear.

-line 275: I would change in: ‘Among these, R. rickettsii and R. parkeri are recognized as pathogenic’ [the others are not]. ‘Exposure to other species is occurring, and could influence the epidemiology and clinical outcomes of 276 other Rickettsia sp. Infections’ [delete ‘although’, this is not in contrast to the previous sentence]

-line 279-80: this is quite obvious..!

Comments on the Quality of English Language

Several sentences are difficult to understand, and many advebs are not used correcly.

Round 2

Reviewer 2 Report

Comments and Suggestions for Authors

I think the manuscript has improved a lot and reads better; I however encourage the authors to read the manuscript again, carefully, to make concepts clearer and check for inconsistencies and mistakes.

A few suggestions:

Please check what the acronym SFGR stands for and use it consistently thoughout the manuscript.. is it referring to the causative agents (e.g. as reported in line 29 and 55) or to the disease (e.g. as in lines 33, 34, 43)?

Line 47: maybe ‘complementing to’ is better than ‘incorportating into’ here

Line 56: explain the acronym RMSF

Line 56 and elsewhere: do nt start sentences with abbreviations (e.g. not ‘R.’ but ‘Rickettsia’)

Line 279 and following paragraph: Please revise. It is not clear to me. You mean that you expect changes in human exposue to other rickettsia species, and suggest more surveillance? (in humans? In ticks?)

Line 301: revise ‘infecting’.. Is it infesting?

Comments on the Quality of English Language

Please check for long sentences and readability
